# Improving Documentation of Nutritional Care in A Nursing Home: An Evaluation of A Participatory Action Research Project

**DOI:** 10.3390/geriatrics4010029

**Published:** 2019-03-20

**Authors:** Christine Hillestad Hestevik, Ella Heyerdahl, Bjørg Lysne Garaas, Gerd Sylvi Sellevold, Marianne Molin

**Affiliations:** 1Department of Physiotherapy, Faculty of Health Sciences, OsloMet—Oslo Metropolitan University, 0130 Oslo, Norway; 2Bjørknes University College, 0456 Oslo, Norway; ella.heyerdahl@bhioslo.no; 3Cathinka Guldberg Centre, 0456 Oslo, Norway; bjorg.garaas@gmail.com; 4Lovisenberg Diaconal University College, 0456 Oslo, Norway; gerd.sylvi.sellevold@ldh.no; 5Department of Nursing and Health Promotion, Faculty of Health Sciences, OsloMet—Oslo Metropolitan University, Norway and Bjørknes University College, 0456 Oslo, Norway; mmolin@oslomet.no

**Keywords:** nursing home, nutrition, older people, participatory action research, documentation, evaluation

## Abstract

Background: Nursing home patients at nutritional risk are often not identified, nor given entitled nutritional treatment. One approach proven suitable to facilitate change in clinical practise is participatory action research (PAR). This is a process which involves research participants in reflection, planning, action, observation, assessing and re-planning, targeted to bring about change. The aim of the present study was to evaluate whether a PAR project resulted in improved documentation of nutritional care in a nursing home ward. Method and sample: A quantitative evaluation. Documentation of the nutritional information was collected from medical records of residents in a nursing home ward at baseline and five months into the project period. Results: Increased documentation of individual nutritional treatment measures was found from baseline to the follow-up. The number of residents with a nutritional care plan (NCP) also increased significantly. On the other hand, the study identified a significant decrease in the proportion of residents with documented weight and nutritional status. Conclusion: The evaluation found several improvements in the documentation of nutritional care practice in the nursing home ward as a result of the PAR project, indicating that a PAR approach is suitable to bring about change in practice.

## 1. Background

Older persons are generally susceptible to undernutrition [1,2] and prevalence among nursing home residents is reported to be high [3,4,5,6]. There are various causes of undernutrition [1,7], of which disease and frailty are the most common [2]. Many older people also experience impaired sense of taste and smell, lack of appetite, oral and swallowing problems, cognitive impairment and depression, which are all risk factors for undernutrition [1]. A large proportion of residents in nursing homes have dementia [8], and persons with dementia are particularly vulnerable to undernutrition [7,9]. Undernourishment has serious negative implications, such as increased risk for morbidity and mortality, impaired cognitive, physical and social function [1,7]. Hence, undernourishment is linked to reduced quality of life and increased healthcare costs [6,7].

Evidence-based international guidelines to prevent and treat undernutrition have been issued to ensure high quality nutritional care in healthcare institutions. They highlight five recommendations that should be implemented in clinical practice: (1) To identify residents at nutritional risk, (2) to compile a nutritional care plan (NCP) including goals and treatment plan, (3) to provide appropriate nutritional treatment, (4) to monitor, assess and adjust the treatment and (5) to document the nutritional status and treatment in the medical record [2]. 

Staff involved in nutritional screening, assessment and care planning should receive appropriate education and training to make sure the nutritional care is in line with current standards [2]. However, several studies showed that there are several barriers to achieve adequate nutritional practice. Problems such as lack of knowledge about nutrition and how to identify undernutrition, uncertainty about roles and responsibilities and lack of allocated time and resources for nutritional care were reported [10,11,12,13,14].

One approach proven suitable in order to facilitate change in clinical practice is participatory action research (PAR) [15,16,17]. Briefly, PAR is a process that involves research participants in, reflection, planning, action, observation, assessing and re-planning to bring about change [18,19]. In PAR, the participants are co-researchers and define the issues and identify the solutions that make the process of change more meaningful for those involved [18]. Also, local knowledge and beliefs are investigated, which increases the possibility to create relevant knowledge [18]. The knowledge created in this process is the knowledge associated with new skills and practices, representing an approach that can help reducing the gap between research knowledge and practice [18,20].

Given current and future challenges related to undernutrition among nursing home residents, it is of great importance to improve nutritional care practice. Here, an important scope of research should be to identify and test models, work processes and tools providing a basis for sustainable changes in work routines. This study is part of a PAR project where the aim was to develop and improve the quality of nutritional care practice in a nursing home ward, i.e. according to the Norwegian guidelines. The aim of the present study was to evaluate the nutritional practice in the ward before and five months into the PAR project to assess whether the project resulted in improved documentation of nutritional practice.

## 2. Materials and Methods

### 2.1. Study Design

The study is a quantitative evaluation of a PAR project, with a pre-posttest design. The nutritional practice was evaluated by examination of nutritional information extracted from the patient’s medical records prior to and five months after initiation of the PAR project.

### 2.2. The PAR PROJECT

The PAR project was a multidisciplinary collaboration led by representatives from two university colleges and a nursing home ward. The project aimed to improve the participating healthcare providers’ competence on assessing nutritional status, providing quality nutritional care and documenting nutritional information in the medical records. Seven project meetings (lasting 1.5–2 h) were arranged regularly over a five-month period and were led by one teacher from each of the university colleges, one nutritionist and one nurse. The project meetings were dialogue based and alternated between practical experience and theoretical reflection. All participants in the project were co-researchers. The participants were able to acquire personal experience, to reflect upon this experience together with the other participants and to receive feedback and gain different perspectives within the multi-professional team. The participants identified professional, organizational and ethical challenges to the nutritional care practice in the nursing home. These challenges were reflected upon and possible solutions on how to improve the practice were agreed on. In the next sessions, experiences from implementing this in practice were discussed and adjustments were made. The participants also requested training in topics within nutrition where they needed to improve their knowledge, resulting in the project leaders providing lectures in these topics (Table 1).

## 3. Setting

This study was performed in a nursing home ward in one of the larger cities in Norway. The nursing home is a modern 112-bed facility. At the time of the project, the ward had 40 long-stay residents. They were older residents with multiple comorbidities, including dementia, who were dependent on nursing assistance day and night. 

### 3.1. Participants In The PAR Project 

The participants were healthcare providers working in the nursing home ward. At the project baseline, the ward had 31 permanent employees (23 women and eight men), including seven nurses, 14 auxiliary nurses and 10 assistant nurses. A group leader from each unit attended all the project meetings. For the rest of the staff, participation in the project meetings depended on who was on duty. The head nurse, in cooperation with the healthcare providers, assessed who should attend the various meetings. Participation had to be organized this way to give all the healthcare providers the opportunity to participate in some of the meetings, while at the same time making sure enough staff were caring for the residents during the sessions. The nurses were professionally responsible documenting the nutritional assessments and nutritional care in the medical records, for the residents in the ward, in cooperation with the rest of the healthcare providers. The unit group leaders were responsible for communicating information from the project meetings to the non-attending staff in their unit.

### 3.2. Data Collection

Content from the medical records was extracted from the electronic medical record (EMR). Baseline and follow-up data were collected simultaneously. This was possible because all registrations in the medical records were marked with time and date. Seven of the residents had moved in on the ward after the baseline date and, therefore, lacked baseline data. Hence, nutritional documentation was collected from 33 medical records at baseline and 40 medical records at follow-up. Documentation of height, weight, nutritional assessment, dietary intake, requirements and individual nutritional treatment measures from each resident was collected from the medical records. 

## 4. Measurement and Procedures

### 4.1. Quality Indicators

Quality indicators, developed by the Norwegian Directorate of Health [21], were used to measure change in documentation of individual nutritional care in the medical records from baseline to follow-up. Quality indicators are standardised measures that are in line with current best practice recommendations, intended to provide important information to be used in internal quality improvement initiatives. Nursing homes can use the indicators as a tool in order to monitor quality over time and to compare quality with other nursing homes [22]. In Norway, the municipality uses quality indicators to assess the quality of nursing homes and other healthcare institutions [23]. The quality indicators used in this study were: (1) Percentage of residents with registered weight, (2) percentage of residents with registered nutritional assessment, (3) percentage of residents at nutritional risk with registered dietary requirements and intake and (4) percentage of residents at nutritional risk with a nutritional care plan (NCP).

### 4.2. The Nutrition Journal (NJ)

NJ is developed to identify residents at nutritional risk [21] and was the nutritional screening tool used by the nursing home. The instructions accompanying the screening tool, state assessment criteria for the various categories of nutritional status as presented in Table 2. In this study, the term “at nutritional risk” describes residents who have been categorised as at risk of undernutrition or severely undernourished. 

### 4.3. Nutritional Care Plan (NCP)

Documentation of nutritional status, percentage of residents with an NCP, dietary requirements and intake and individual nutritional treatment measures at baseline and follow-up, were collected from the individual NCPs in the medical records. All individuals identified as at nutritional risk should have an appropriate NCP, including the resident’s nutritional status, dietary intake and requirements and individual nutritional treatment measures. Information about adjustment of physical, psychological and social factors impeding adequate dietary intake should also be included [2]. Commercially produced supplements, like nutritional drinks, and vitamin/mineral supplements are important measures in the individual nutritional treatment of residents at nutritional risk, but these were excluded from this study since they were documented as medication in the medical record and not as individual nutritional treatment measures in the NCPs. 

## 5. Data Analyses

IBM SPSS version 22 for Windows was used for statistical analysis. Patient characteristics and documentation of nutritional care in medical records at baseline and follow up are presented as frequencies and percentages. Due to the small sample in this study, nonparametric statistics were used. Categorical variables were compared by Mc Nemars test. *p*-values < 0.05 were considered statistically significant. The SPSS dataset used for analysis is provided as Appendix A.

## 6. Ethics

The main project was approved by Norwegian Social Science Data services (NSD). Only readily available resident administrative data were examined. The medical records were printed out in the nursing home and anonymised before data was extracted.

## 7. Results

### 7.1. Resident Characteristics

The majority of the residents were women (78%) and the average age of the residents was 85.5 (±9.1) years. The proportion of residents with documented undernourishment or risk of undernourishment was 36.4% at baseline and 27.5% at follow-up (Table 3). 

### 7.2. Documentation of Individual Nutritional Care

The proportion of all residents with an NCP increased significantly from baseline to follow-up (Table 4). Also, the proportion with registered dietary intake and registered dietary requirement calculation increased, although not significantly (Table 4). However, the proportion of residents with recorded weight and nutritional status assessment decreased significantly from baseline to follow-up.

### 7.3. Individual Nutritional Treatment Measures

In the NCPs, documentation of all included individual nutritional treatment increased between baseline and follow-up, most of them significantly. These included documentation of food preferences, which increased from 18.2% (SD 39.2) to 42.5% (SD 50.1), *p* < 0.01, and facilitation of psychosocial environment, which increased from 45.5% (SD 50.6) to 65.0% (SD 48.3), *p* = 0.02. Also, practical arrangements before and during meals increased from 39.4% (SD 49.6) to 70.0% (SD 46.4), *p* < 0.01, as well as documentation of customized food, including fortification of food, which increased from 15.2% (SD 36.4) to 32.5% (SD 47.4), *p* = 0.03 between baseline and follow-up. In addition, snacks between meals increased, although not significantly, from 9.1% (SD 29.2) to 17.5% (SD 38.5). 

## 8. Discussion

### 8.1. Main Findings

The present study found that using a PAR approach improved nutritional care practice in a nursing home in some areas, but not in others. Measures indicating improved care include a significantly increased proportion of residents with an NCP identified at follow-up compared to baseline and a significant increase in the documentation of various individual treatment measures. Additionally, the results identified a significant decrease in the number of residents with a recorded weight and nutritional status assessment between baseline and follow-up.

The number of residents with an NCP increased and 75 % of the residents at nutritional risk had such a plan at the follow-up. The guidelines state that all residents at nutritional risk are entitled an NCP [2]. However, most of the residents had an NCP irrespective of nutritional status, which may result in less attention given to residents at nutritional risk. On the other hand, considering the poor health of nursing home residents and their vulnerability to under- and/or malnutrition [6,24], most nursing home residents will require individual arrangements in relation to their meals, and such information should therefore be recorded in an NCP [2]. The increase in residents with an NCP indicate an overall increased focus on individual nutritional measures among the staff. 

For residents at nutritional risk, dietary requirements and dietary intake should be specified in the NCP. Moreover, it is important that the dietary intake is assessed in conjunction with the dietary requirement to be able to identify whether or not the patient has a satisfactory dietary intake [2]. Although this study shows a small increase in the documentation of dietary intake, the majority of the NCPs of residents at nutritional risk lacked this information. Similarly, very few NCPs included estimations of the patient’s dietary requirements. Studies show that healthcare providers often lack knowledge about how to register dietary intake and calculate dietary requirements [25,26], possibly also explaining the lack of improvement in this study. Although this information was a selected topic in the dialogue-based teaching sessions, it may not have been communicated well enough to non-attending staff. One study found that health care professionals in nursing homes seemed to lack clear directions as to what nutritional information to document in the medical record [11]. Also, the healthcare providers might have had to prioritize more urgent tasks due to lack of time caused by challenges, like of shortage of staff and heavy work load. Many European countries have reported shortage of healthcare providers, poor working conditions and high turnover among staff as significant difficulties in the long-term care sector [27], and these challenges are similar in Norway [28,29]. Furthermore, to reflect upon practice takes time, accordingly it takes time to learn new ways of doing things [30]. With this in mind, the short time line between the baseline and follow-up measure in this study may also explain the lack of improvement in some areas.

Surprisingly, the results show that the proportion of residents whose weight and nutritional status assessment were documented declined from baseline to follow-up. This is alarming, since documentation of nutritional assessment and weight history in the medical record is an important part of adequate nutritional care [2]. Unfortunately, research shows that lack of information and inaccurate documentation in the medical record is a challenge in nursing homes [31,32]. Adequate documentation of nutritional care is demanding in situations where there are limited resources in addition to lack of personnel with the necessary competence [33,34]. Additionally, the EMR system often has poor functionality, making appropriate documentation of nutritional care difficult [35,36]. This was also reported to be the case for the healthcare providers in the present study, possible resulting in inadequate documentation. Although auxiliary nurses and assistant nurses contributed to weighing and assessing nutritional status, only the nurses were responsible for recording this information in NJ, possibly making this part of the nutritional practice more vulnerable to shortages of staff and heavy workloads. 

There are high expectations of standards of nutritional care practice in nursing homes [2,9], but studies show that institutional constraints can make it difficult to pursue the right course of action [25,27,37]. Proper documentation, in this case of nutritional care, in the medical record is required to ensure that personnel have sufficient information to be able to provide appropriate care and treatment to the residents [11,36]. Healthcare institutions are obliged to make this achievable, enabling the staff to meet the requirements for professional conduct [2,38,39]. 

The results from this study indicate that a PAR approach in form of dialogue-based teaching sessions significantly increased the documentation of individual customized nutrition care. This indicates that the participants managed to convert the practical and theoretical knowledge they gained through the dialogue based project meetings into their daily clinical practice and that they developed their skills and practices. A prerequisite for a successful PAR project is that the participants experience the gained knowledge as valuable and relevant [40]. Thus, it is possible that the participants here considered improvement of individual environmental and practical facilitation in connection with meals being more valuable for the residents than some of the other objectives of the PAR that did not improve or only improved slightly. It may be that this area of the nutritional practice gave the healthcare providers a greater feeling of interacting with the residents, thus meeting their needs. In addition, this aspect of the nutritional practice may have been emphasized more in the dialogue-based teaching sessions compared to other activities, possibly contributing to the positive changes in this particular area. Another possible explanation is that the participant’s involvement and influence to make changes to the daily routines was greater in this area of the nutritional practice than others. 

### 8.2. Methodical Considerations

An important part of action research is to evaluate how the research affects practice [41]. In this study, change in the nutritional practice was evaluated by examining medical records. This source of data was chosen since documentation of nutritional care is an important part of nutritional practice [21,39], and several of the variables used in this study are national quality indicators which municipalities in Norway use when they assess quality of nutritional care in nursing homes [23]. These national quality indicators are knowledge-based and relevant when measuring nutritional care practice [42] and they may help pinpoint how to improve quality of care [22]. However, they are often criticised as representing measures of documentation rather than actual care, and the lack of improvement seen in some of the measures may therefore be explained by poor documentation rather than lack of nutritional care to the residents. In evaluation of quality there are also concerns about validity and reliability of these measures and that there exists numerous definitions of quality [22]. The PAR project was also evaluated in a qualitative study (preliminary unpublished data). 

Collecting data from medical records made it possible to measure practice without much external interference. Using this method, the results may give a good indication of how a PAR approach may influence healthcare practice in real everyday settings. The results from this evaluation were presented to the participants during the project. This was a strength as it helped emphasize problem areas in the nutritional care practice in the ward. 

The fact that the head nurse was responsible for organizing participants’ attendance may have influenced the results as this may have affected the participants’commitment to the project. Some may have felt pressured to participate or wanted to participate but were not allowed. For a PAR approach to be successful, the necessary framework and conditions need to be present. It is therefore important to evaluate whether the institution and the participants at all levels are committed and agree to participate actively in the project [40]. Likewise, the external researcher need to have an empathetic understanding of the everyday challenges that staff face and adapt the project activities accordingly since lack of support from the project leaders can slow or hinder the project [18]. Although the preconditions for a successful project seemed to be present in this study, unforeseen challenges, such as absence of staff due to sick-leave and challenges in communicating information to non-attending staff, were encountered along the way. 

The absence of information on documented use of nutritional supplements is a limitation to this study. The use of oral nutritional supplements is often an important part of nutritional care and is recommended when normal diet is insufficient to meet daily nutritional requirements. The use of such supplements is associated with positive outcomes, such as increased energy intake and body weight [2].

## 9. Conclusions

In conclusion, this study showed that using a PAR approach, resulted in positive changes in the documentation of nutritional practice, indicating that a PAR approach is suitable to bring about practice change in a nursing home. To be able to achieve improvements in nutritional practice it is essential to engage staff and leaders in particular, in order to maintain focus over time and thereby make sustainable improvements in the nutritional care practice.

## Figures and Tables

**Table 1 geriatrics-04-00029-t001:** Topics of the dialogue-based teaching sessions in the PAR project.

	Teaching Topics That Were Requested By The Participants
Session 1	How to identify residents at nutritional risk How to calculate dietary requirements How to provide residents at nutritional risk with appropriate nutritional treatment
Session 2	Laws and guidelines related to the documentation of nutritional practice Documentation of nutritional practice in the medical record Basic nutritional knowledge Calculation of dietary requirements and registration of dietary intake
Session 3	Individual NCP Documentation of nutritional practice in the medical record Different screening tools - student experiences
Session 4	Individual nutritional treatment measures and documentation of these in the individual NCP Nutritional supplements
Session 5	Accurate assessment of nutritional status Accurate measurement of height and weight Accurate documentation of dietary intake Documentation of nutritional treatment measures in the NCP
Session 6	Assessment of nutritional status Documentation of nutritional treatment measures in the NCP
Session 7	Laws and guidelines related to documentation of nutritional practice Ways to fortify food to increase energy content

**Table 2 geriatrics-04-00029-t002:** Assessment criteria for the various nutritional status categories as specified in “The Nutrition Journal” (NJ).

Good Nutritional Status	Risk of Undernutrition	Severely Undernourished
Requires:	May be present if one or more of the following nutritional risk indicators are present:	Requires presence of the following nutrition risk indicators:
Normal dietary intake Normal BMI No weight loss No clinical signs of undernutrition	Reduced dietary intake BMI below 22 (>65 years) Weight loss up to 5% in the last two months or up to 10% in the last six months Presence of one or more nutrition-related problems	Reduced dietary intake BMI below 22 (>65 years) Weight loss of >5% in the last two months or >10% in the last six months Visible clinical signs of undernutrition

**Table 3 geriatrics-04-00029-t003:** Documented nutritional status in the medical records at baseline (n = 33) and follow-up (n = 40).

Nutritional Status	Baseline	Follow-Up
	Frequency (%)	Frequency (%)
Good nutritional Status	20 (60.6)	16 (40.0)
At risk of undernutrition	9 (27.3)	7 (17.5)
Severely undernourished	3 (9.1)	4 (10.0)
Nutritional status not recorded	1 (3.0)	13 (32.5)

**Table 4 geriatrics-04-00029-t004:** Quality indicators for nutritional care developed by the Norwegian Directorate of Health [10]. The proportion (%) and standard deviation (SD) at baseline and follow-up are given.

Quality Indicator.	Baseline % (SD)	Follow-Up % (SD)	*p*-value ^1^
Weight (all residents ^2^)	84.8 (36.4)	63.6 (47.4)	0.039
Nutritional status assessment (all residents)	97.0 (17.4)	67.5 (47.4)	0.008
NCP (residents at nutritional risk or severely malnourished ^3^)	25.0 (45.2)	75.0 (45.2)	0.031
NCP (all residents)	30.3 (46.7)	67.5 (47.4)	0.000
Registered dietary intake (residents at nutritional risk or severely malnourished)	0.0 (0.0)	25.0 (45.2)	0.250
Registered dietary intake (all residents)	9.1 (29.2)	17.5 (38.5)	0.250
Registered dietary requirement calculation (residents at nutritional risk or severely malnourished)	0.0 (0.0)	8.3 (28.9)	1.00
Registered dietary requirement calculation (all residents)	0.0 (0.0)	2.5 (15.8)	1.00

^1^ Mc Nemars test, ^2^ n = 33; only residents with registered quality indicators at both baseline and follow-up are included, ^3^ n =12; only residents with registered quality indicators at both baseline and follow-up that were at nutritional risk or severely malnourished are included.

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
