# Peer review of "Improving Documentation of Nutritional Care in A Nursing Home: An Evaluation of A Participatory Action Research Project"

_geriatrics, 2019, doi:10.3390/geriatrics4010029_

Round 1

Reviewer 1 Report

Thank you for the opportunity to review the manuscript: "Improving Documentation of Nutritional Care in A 2 Nursing Home: An Evaluation Of A Participatory 3 Action Research Project" The topic is significant for health care professionals and could have several consequences for patients.  

The paper itself sounds interesting, but needs to be extensively reviewed. It's quite confusing to find data in the discussion section, not analysed in the previous ones and even not related to the background.

References are quite aged, and in particular, some of them have been revised or updated by more recent studies

background, row 38 authors cited aged references, and there are more recent studies affirming what they are explaining, maybe those could be updated; row 50, reference 2 is related to the 2019 guidelines, why Authors also cited the previous ones?

Methods: 

The study design seems an observational,  longitudinal pre-post quantitative study.

Setting: Authors underline care traditions about the cited nursing home, this paragraph needs more explanation to contextualise what they mean with this sentence.

Participants: row 109-111, where did nurses document nutritional records? In the standard clinical record or a specific form?

Data collection: I cannot understand why patients in the follow-up group were more than the baseline ones. Can the authors explain this difference?

Row 121: the "nutrition Journal cited in this section, but explained in the following results quite confounding. Please explain.

Table 3: is not clear: put indicators over the number or following them. (i.e. "%" have been put un the title, first column) Moreover, give some explanation about the distribution of the percentage (i.e. standard deviation, confidence limits).

Documentation, row 173, Authors describe 8 more patients in the follow-up group, do they influence the results?

Discussion: authors cited nurses shortage and lack of staffing, but they do not consider this evidence in the background section, Do they find something in data analysis? Need an explanation on what they correlate with staffing.

Methodical consideration: Authors are explaining results that appear only in the discussion section. Please explain.

Author Response

Comments and Suggestions for Authors

Thank you for the opportunity to review the manuscript: "Improving Documentation of Nutritional Care in A Nursing Home: An Evaluation Of A Participatory Action Research Project". The topic is significant for health care professionals and could have several consequences for patients. 

The paper itself sounds interesting, but needs to be extensively reviewed.

o   It's quite confusing to find data in the discussion section, not analysed in the previous ones and even not related to the background.

Thank you for this valuable comment. We are here referring to findings of a qualitative study that explores the participant’s experiences from the project. However, since these results are not yet published, we have decided to delete these parts of the discussion.

o   References are quite aged, and in particular, some of them have been revised or updated by more recent studies. background, row 38 authors cited aged references, and there are more recent studies affirming what they are explaining, maybe those could be updated; row 50, reference 2 is related to the 2019 guidelines, why Authors also cited the previous ones?

Thank you for this comment. We have updated the references in general were possible and have deleted citations to old guidelines.

Methods:

The study design seems an observational, longitudinal pre-post quantitative study.

o   Setting: Authors underline care traditions about the cited nursing home, this paragraph needs more explanation to contextualise what they mean with this sentence.

The diaconal tradition, that the nursing homes caring philosophy is rooted in, is a two thousand year old Christian tradition. The core values is to meet each patient and their next of kin with respect and to speak out for the vulnerable groups in society.  We have however realized that this information is not relevant to this study and decided to delete this information from the setting section on page 3.

o   Participants: row 109-111, where did nurses document nutritional records? In the standard clinical record or a specific form?

The nurses documented everything in medical records, but the nutritional assessment was done using a specific form. The results from this assessment was then documented in the medical record. We have tried to make this clearer by writing that data was collected from the residents’ medical records on page 3 line 111 and page 4, line 124.

o   Data collection: I cannot understand why patients in the follow-up group were more than the baseline ones. Can the authors explain this difference?

We agree that this was poorly explained. We collected baseline and follow-up data simultaneously. This was possible because all registrations in the medical records are dated.  We registered what was documented at time of baseline and at time of follow-up. Seven of the residents on the ward had moved in after the baseline date and therefor it was not possible to collect baseline data for these residents. We have now elaborated on this on page 3-4, line 117-121.

o   Row 121: the "nutrition Journal cited in this section, but explained in the following results quite confounding. Please explain.

The results from the nutritional assessments from ‘the nutritional Journal’ was registered in the medical record and we have therefore corrected this to “measures from each resident was collected from the medical records” on page 4, line 124.

o   Table 3: is not clear: put indicators over the number or following them. (i.e. "%" have been put under the title, first column) Moreover, give some explanation about the distribution of the percentage (i.e. standard deviation, confidence limits).

We have edited this table to make it easier to read. In addition we have added information about standard deviation to all the results.

o   Documentation, row 173, Authors describe 8 more patients in the follow-up group, do they influence the results?

Thank you for this comment. It would of course be better if we had baseline and follow up data for all 40 residents as this would have given a larger sample size. We do however not think this is crucial for the results as we measure level of documentation in %. When preforming the Mc Nemars test, only residents with registered quality indicators at both baseline and follow-up were included.

o   Discussion: authors cited nurses shortage and lack of staffing, but they do not consider this evidence in the background section, Do they find something in data analysis? Need an explanation on what they correlate with staffing.

Methodical consideration: Authors are explaining results that appear only in the discussion section. Please explain.

Thank you for making us aware of this and we do agree that this seems confusing. We are here referring to findings of a qualitative study of the participants experiences from the project. Since these results are not published yet, we have decided to delete these parts of the discussion.

Reviewer 2 Report

The manuscript presents a pre- and post-analysis of changes in the rates of nutritional documentation by healthcare professionals within a nursing home environment after taking part in a participatory action project; the project included seven lessons aimed at enhancing the professionals’ knowledge about nutritional guidelines. At 5-month follow-up, the authors found an increase in certain measures of quality of nutritional care, but not in others.

The paper is interesting and well-presented. It addresses a topic of interest, as quality of nutritional status in long-term care settings for older people is an ongoing issue. What the findings indicate is a change in documentation routines rather than quality of nutritional care itself (which can be inferred as benefiting from more adequate documentation), but the authors present a satisfactory discussion of this point in the Discussion section. A number of intervening factors are identified and discussed by the authors. The rationale for the study is well documented and the conclusions are supported by the results.

I have some minor comments that I would like the authors to address:

1)      Introduction, p.2 line 45 – Please clarify whether these guidelines are national or international

2)      Introduction, p.2 line 52 – Small typo: “nutritional care are” should be “nutritional care is”

3)      Introduction, p.2 line55 - Small typo: “recourses” should read “resources.

4)      Methods, p.2 line 85 – Small typo: “altered” should read “alternated”

5)      Methods, section 2.2 – Please clarify that the participants were healthcare professionals. I know that you have a dedicated section about this  (3.1) on the next page, but even just mentioning it at the beginning of the methods helps the reader.

6)      Section 3.2 Data collection – I am bit confused by your baseline and follow-up records numbers, which are indicated to be 33 and 40 respectively. Considering the loss of residents between the two time points described in this section, one would expect that there were fewer records at follow-up than baseline. It would be useful to clarify this.

7)      Nutritional care plan, p.5 lines 152-155 – I feel that the absence of information on supplements is a limitation for this study and should be mentioned in the discussion.

8)      Table 4, pp.5-6 – This table should present effect sizes, which are usually calculated as Cohen’s g (1988). Given the small sample size, effect sizes can give a clearer understanding of the magnitude of the change.

9)      Table 4, pp.5-6 – The rows appear to be misaligned. Please fix this to ensure readability.  

Author Response

Reviewer 2: Comments and Suggestions for Authors

The manuscript presents a pre- and post-analysis of changes in the rates of nutritional documentation by healthcare professionals within a nursing home environment after taking part in a participatory action project; the project included seven lessons aimed at enhancing the professionals’ knowledge about nutritional guidelines. At 5-month follow-up, the authors found an increase in certain measures of quality of nutritional care, but not in others.

The paper is interesting and well-presented. It addresses a topic of interest, as quality of nutritional status in long-term care settings for older people is an ongoing issue. What the findings indicate is a change in documentation routines rather than quality of nutritional care itself (which can be inferred as benefiting from more adequate documentation), but the authors present a satisfactory discussion of this point in the Discussion section. A number of intervening factors are identified and discussed by the authors. The rationale for the study is well documented and the conclusions are supported by the results.

I have some minor comments that I would like the authors to address:

1)       Introduction, p.2 line 45 – Please clarify whether these guidelines are national or international

These are international guidelines. We have clarified this in the text on page 2 line, 45.

2)       Introduction, p.2 line 52 – Small typo: “nutritional care are” should be “nutritional care is”

3)       Introduction, p.2 line55 - Small typo: “recourses” should read “resources.

4)       Methods, p.2 line 85 – Small typo: “altered” should read “alternated”

Thank you for these comments. We have corrected the typos.

5)       Methods, section 2.2 – Please clarify that the participants were healthcare professionals. I know that you have a dedicated section about this  (3.1) on the next page, but even just mentioning it at the beginning of the methods helps the reader.

Thank you for this comment. We added information about the participants being healthcare providers in the beginning of The PAR Project section on page 2, line 81-82. We call the participants healthcare providers throughout the text.

6)       Section 3.2 Data collection – I am bit confused by your baseline and follow-up records numbers, which are indicated to be 33 and 40 respectively. Considering the loss of residents between the two time points described in this section, one would expect that there were fewer records at follow-up than baseline. It would be useful to clarify this.

Thank you for this valuable comment. We agree that this is not well explained. We collected baseline and follow-up data simultaneously. This was possible because all registrations in the medical records are dated.  We registered what was documented at time of baseline and at time of follow-up. Seven of the residents on the ward had moved in after the baseline date and therefore it was not possible to collect baseline data for these residents. We have now elaborated on this on page 3-4, line 117-121.

7)       Nutritional care plan, p.5 lines 152-155 – I feel that the absence of information on supplements is a limitation for this study and should be mentioned in the discussion.

This limitation is now mentioned at the end of the discussion on page 8-9, line 300-304.

8)       Table 4, pp.5-6 – This table should present effect sizes, which are usually calculated as Cohen’s g (1988). Given the small sample size, effect sizes can give a clearer understanding of the magnitude of the change.

Thank you for this comment. We do agree that more information about spread and variation of the data is useful, hence we have added the standard deviations in the table. Additionally, the Mc Nemar test is a significance test to explore changes in proportions of paired data. The magnitude of changes can be indicated by the p-values of the test, and we have therefore added these in the table.

9)       Table 4, pp.5-6 – The rows appear to be misaligned. Please fix this to ensure readability. 

We have made corrections to make the tables more presentable.

Round 2

Reviewer 1 Report

Thank you to give me the opportunity to evaluate the revised form of the paper.

Authors improve significantly the paper and allow it's readability.

all the suggestions proposed have been fulfilled.